# Whole genome sequencing identifies genetic variants associated with neurogenic inflammation in rosacea

Zhili Deng [1,2,3,7], Mengting Chen [1,2,3,7], Zhixiang Zhao[1,2,3], Wenqin Xiao[1,2,3], Tangxiele Liu[1,2,3], Qinqin Peng[1,2,3], Zheng Wu[1,2,3], San Xu[1,2,3], Wei Shi[1,2,3], Dan Jian[1,2,3], Ben Wang[1,2,3], Fangfen Liu[1,2,3], Yan Tang[1,2,3], Yingxue Huang[1,2,3], Yiya Zhang[1,2,3], Qian Wang[4], Lunquan Sun [3,5], Hongfu Xie[1,2,3], Guohong Zhang [6] ✉ & Ji Li [1,2,3] ✉

Rosacea is a chronic inflammatory skin disorder with high incidence rate. Although genetic predisposition to rosacea is suggested by existing evidence, the genetic basis remains largely unknown. Here we present the integrated results of whole genome sequencing (WGS) in 3 large rosacea families and whole exome sequencing (WES) in 49 additional validation families. We identify single rare deleterious variants of *LRRC4*, *SH3PXD2A* and *SLC26A8* in large families, respectively. The relevance of *SH3PXD2A*, *SLC26A8* and *LRR* family genes in rosacea predisposition is underscored by presence of additional variants in independent families. Gene ontology analysis suggests that these genes encode proteins taking part in neural synaptic processes and cell adhesion. In vitro functional analysis shows that mutations in *LRRC4*, *SH3PXD2A* and *SLC26A8* induce the production of vasoactive neuropeptides in human neural cells. In a mouse model recapitulating a recurrent *Lrrc4* mutation from human patients, we find rosacea-like skin inflammation, underpinned by excessive vasoactive intestinal peptide (VIP) release by peripheral neurons. These findings strongly support familial inheritance and neurogenic inflammation in rosacea development and provide mechanistic insight into the etiopathogenesis of the condition.

Rosacea is a common chronic inflammatory skin disorder characterized by flushing, transient or persistent erythema, telangiectasia, papules/pustules, hyperplasia, or a combination of these manifestations, which typically involves the central face[1,2]. The prevalence of rosacea was estimated to be about 5.46% of the global adult population and 3.48% of the Chinese population[3,4]. Currently, although the pathogenesis of rosacea is not fully understood, diverse environmental and endogenous factors have been shown to stimulate a neuromodulator-mediated augmentation of neurovascular dysregulation and abnormal immune responses[5–7].

As an endogenous factor, genetic component might contribute to the development of rosacea, supported by the fact that the prevalence is highest in Celtic or Northern European descendants, and population with fair skin is more likely to be involved, but

[1]Department of Dermatology, Xiangya Hospital, Central South University, Changsha, Hunan, China. [2]Hunan Key Laboratory of Aging Biology, Xiangya Hospital, Central South University, Changsha, Hunan, China. [3]National Clinical Research Center for Geriatric Disorders, Xiangya Hospital, Central South University, Changsha, Hunan, China. [4]Hunan Binsis Biotechnology Co., Ltd, Changsha, Hunan, China. [5]Key Laboratory of Molecular Radiation Oncology Hunan Province, Changsha, China. [6]Department of Pathology, Shantou University Medical College, Shantou, China. [7]These authors contributed equally: Zhili Deng, Mengting Chen. ✉e-mail: g_ghzhang@stu.edu.cn; liji_xy@csu.edu.cn

African Americans and Asians are less affected[1,8]. Previous studies have generated initial insights into the genetic architecture, and identified several common variants associated with rosacea in sporadic case-control datasets via population-based genome-wide association study (GWAS)[9,10]. However, for most complex diseases, common variants only minimally contribute to disease development. Instead, growing evidence have suggested that rare genetic variants in different genes that could not be captured by GWAS, might be more important than common variants in the

susceptibility of diseases[11,12]. Family history is up to 30%[4,13] and positively associated with the risk of rosacea[14,15], which suggests a familial inheritance and provides a valuable resource for the genetic study of this disorder. With the advance of high-throughput sequencing technologies, whole-genome sequencing (WGS) or whole-exome sequencing (WES), especially disease family-based WGS or WES, provides an elegant platform to seek rare susceptibility or pathogenic variants[16–18]. Such studies, however, have not yet been reported in the field of rosacea.

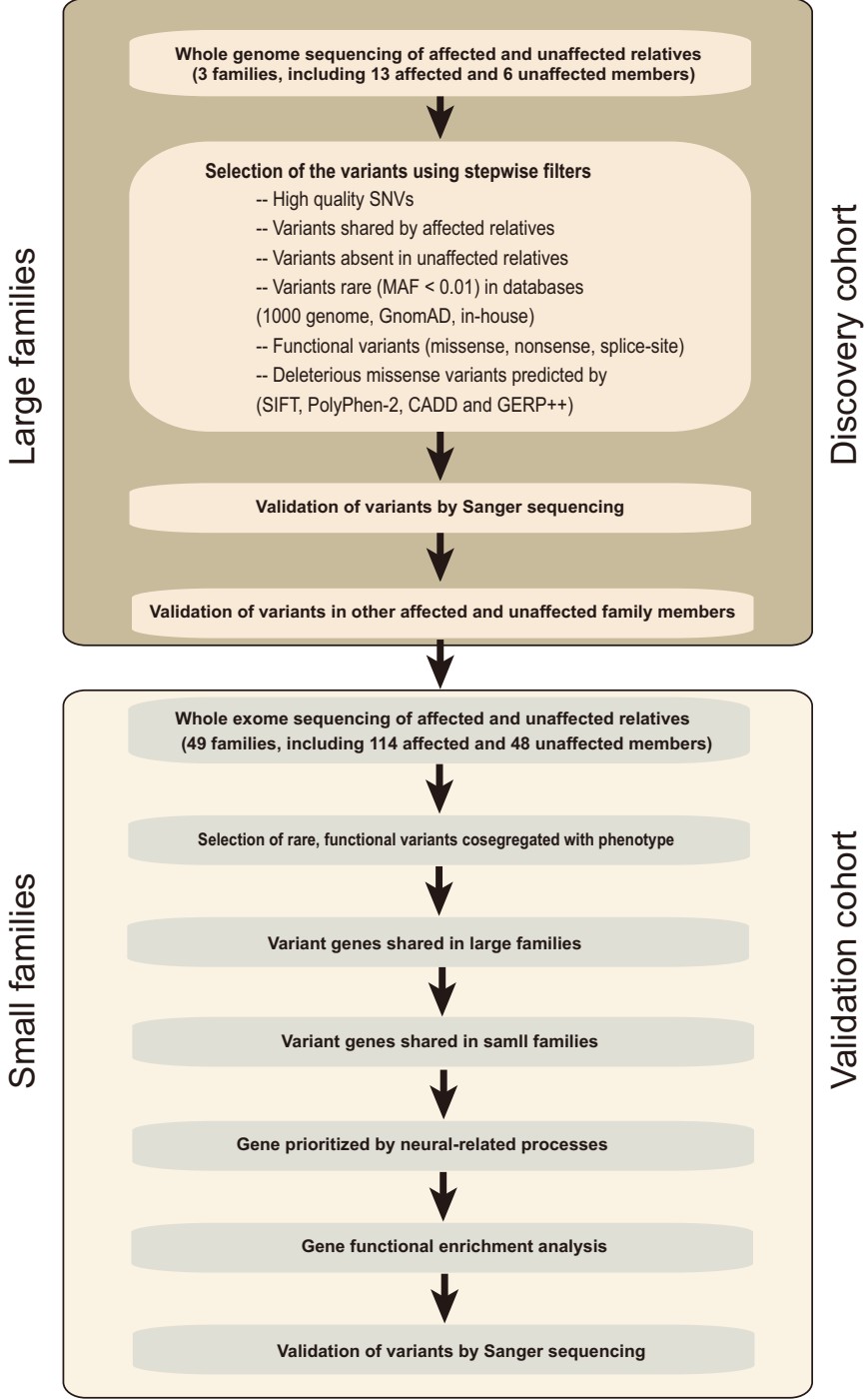

**Fig. 1 | Overview of the study design for sequencing data analysis.** Flowchart of the variant filtering strategy for whole-genome sequencing in multiplex large families, and candidate gene prioritization and gene set enrichment of whole-exome sequencing in small families. MAF minor allele frequency, GnomAD Genome Aggregation Database, SIFT Sorting Intolerant From Tolerant, Polyphen2 Polymorphism Phenotyping v2, CADD Combined Annotation-Dependent Depletion, GERP++ Genome Evolutionary Rate Profiling version 2.

In this study, via WGS and WES analysis of samples from members in rosacea families, we reveal the substantial genetic heterogeneity, and the genetic basis of familial inheritance in rosacea. Combining the use of in vitro and in vivo models, we further demonstrate the important regulatory role of these genetic variants in the neurogenic inflammation in this disease with complex etiopathogenesis.

## Results

### Single rare deleterious variants were identified in the discovery cohort via whole-genome sequencing

To investigate the rare genetic variants associated with rosacea pathogenesis, we performed WGS. After sample quality control (shown in "Methods"), WGS data from three large rosacea families (including 13 patients and 6 healthy individuals), as discovery cohort, were used for subsequent analyses. The overview of the study design for sequencing data analysis was shown in Fig. 1. After calling of variants, filtering, and initial co-segregation analysis within families, we identified rare (minor allele frequency of <0.01), heterozygote, single deleterious variants (SNVs) respectively in the genes of *LRRC4* (c.T1157C, p.L386P) in family 1, *SH3PXD2A* (c.C2281T, p.R761C), marker of proliferation Ki-67 (*MKI67*, c.401-1 G > A) and lysine methyltransferase 2C (*KMT2C*, c.C13522A, p.P4508T) in family 2, innate immunity activator (*C1orf106*, c.G1055C, p.C352S) and *SLC26A8* (c.G1876A, p.V626I) in family 3 (Table 1 and Supplementary Data 1). These rare variants in *LRRC4*, *SH3PXD2A,* and *SLC26A8* were further verified by extended co-segregation analysis via Sanger sequencing (Fig. 2). Moreover, we found that the amino acid residues corresponding to these variant sites are all highly conserved across species, and the variant amino acid residue in *LRRC4* located in the immunoglobulin (Ig)-like domain (Supplementary Fig. 1). Taken together, these results suggest that *LRRC4*, *SH3PXD2A,* and *SLC26A8* each with an identified rare variant are most likely candidate susceptibility genes associated with rosacea, but there exists a high genetic heterogeneity since no single proposed candidate disease-causing gene is identified across all three large families.

### Rare deleterious variants in *SH3PXD2A*, *SLC26A8,* and *LRR* family genes were replicated in the validation cohort via whole-exome sequencing

To substantiate the notion that our results regarding the candidate genes identified in discovery cohort, might have an application to other independent families, settled as validation cohort, total 49 additional families with multiply affected members had been collected and performed with WES (Fig. 1 and Supplementary Fig. 2). We identified a series of rare, heterozygote variants in these validation rosacea families (Supplementary Data 2). Thereinto, additional variants in *SH3PXD2A* (c.G1606A, p.G536S) and *SLC26A8* (c.T772C, p.S258P) were found in family 312 and family 319, respectively, which were further validated by Sanger sequencing (Table 2 and Fig. 3a). Although there was no *LRRC4* variant found in validation families, splice site and missense variants in multiple *LRR* family genes were identified. Specifically, a missense variant in *LRRC43* (c.T1472C, p.V491A) in family 42, a splicing site in *LRRC44* (*c.708-2 A > G*) in family 332, a missense variant in *LRRC47* (c.A1282C, p.K428Q) in family 94, a missense variant in *LRRC55* (c.C292T, p.R98W) in family 313, a missense variant in *LRRCC1* (c.G1931A, p.R644H) in family 147, a stopgain variant in *LRRCC1* (c.C2700G, p.Y900X) in family 147, a stopgain variant in *LRRD1* (c.C2046G, p.Y682X) in family 311, the same missense variant in *LRRD1* (c.A2173G, p.I725V) in family 314 and 399, a missense variant in *LRRTM4* (c.A379G, p.N127D) in family 48 were found (Table 2 and Supplementary Data 2). Collectively, these data further highlight *SH3PXD2A*, *SLC26A8* and *LRR* family genes as potential candidate genes for rosacea susceptibility.

**Table 1 | Candidate genes identified in large rosacea families via WGS**

| Family | Gene | Genomic position | Reference/alterative | cDNA change | Consequence | Amino acid change | GnomAD MAF | SIFT | Polyphen2 |
|---|---|---|---|---|---|---|---|---|---|
| 1 | LRRC4 | 7: 127669537 | A/G | c.T1157C | Missense | p.L386P | 0.000008244 | 0.025, Deleterious | 0.980, Damaging |
| 2 | SH3PXD2A | 10: 105362610 | G/A | c.C2281T | Missense | p.R761C | 0.0000907 | 0.000, Deleterious | 0.997, Damaging |
| 2 | MKI67 | 10: 129914272 | C/T | c.401-1 G > A | Splicing | NA | 0.0003 | NA | NA |
| 2 | KMT2C | 7: 151845490 | G/T | c.C13522A | Missense | p.P4508T | 0.00003295 | 0.041, Deleterious | 0.999, Damaging |
| 3 | SLC26A8 | 6: 35922970 | C/T | c.G1876A | Missense | p.V626I | 0.0004 | 0.007, Deleterious | 0.998, Damaging |
| 3 | C1orf106 | 1: 200880676 | G/C | c.G1055C | Missense | p.C352S | 0.00002488 | 0.018, Deleterious | 0.987, Damaging |

GnomAD Genome Aggregation Database, *MAF* minor allele frequency, *SIFT* Sorting Intolerant From Tolerant, *Polyphen2* Polymorphism Phenotyping v2, *NA* not applicable.

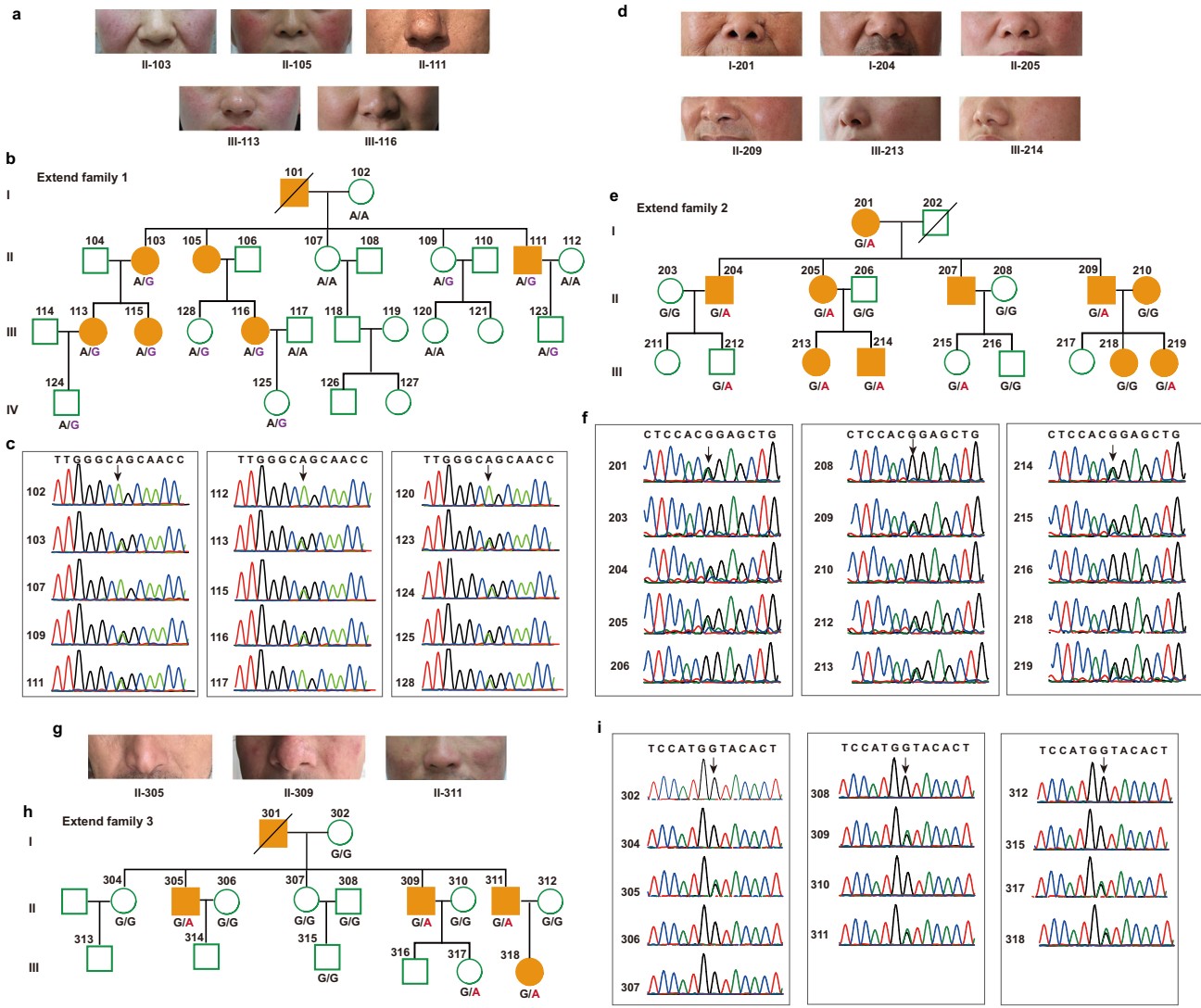

**Fig. 2 | Single rare deleterious variants are identified in large rosacea families. a–c** Images show the individuals with typical rosacea phenotypes in their central faces in large family 1 (**a**). Pedigree structure of the large family 1 (**b**). Solid symbols indicate individuals affected with rosacea; open symbols denote unaffected relatives; squares indicate male individuals; circles denote female individuals and slashes show deceased members. Chromatograms of Sanger sequencing show the heterozygous mutation in *LRRC4* in large family 1 (**c**). **d–f** Images show the individuals affected with rosacea (**d**), pedigree structure (**e**), and Sanger sequencing chromatograms show the heterozygous mutation in *SH3PXD2A* (**f**) in large family 2. **g–i** Images show the individuals affected with rosacea (**g**), pedigree structure (**h**), and Sanger sequencing chromatograms show the heterozygous mutation in *SLC26A8* (**i**) in large family 3.

## Functional analyses of variant genes highlight neural function

Despite increasing evidence suggesting rosacea is a kind of neurogenic skin inflammation[5,6,19,20], it is unclear whether the genetic elements are involved in this neurogenic process. To find the potential genetic variants responsible for neurogenic inflammation in rosacea, we prioritized the genetic variants in each family and ranked the variant genes according to the biological function in neural process. Total 28 genes harboring rare segregating variants were identified in 33 families, including *LRRC4, SH3PXD2A*, and *SLC26A8* in large rosacea families (Fig. 3b). Notably, multiple variants (c.A1195C, p.T399 in family 209, c.C1966G, p.P656A in family 213 and c.A880T, p.I294F in family 313) were identified in *PCDHA5*, a neural cadherin-like cell adhesion gene, in three independent families (Fig. 3b and Supplementary Data 2). To further dissect the shared genetic spectrum across families, we performed gene ontology (GO) and Kyoto Encyclopedia of Genes and Genomes (KEGG) analyses with these variant genes. Interestingly, the genes appeared to cluster within peptidyl-tyrosine dephosphorylation, neural synaptic function and cell adhesion, long-term depression, and neuroactive ligand-receptor interaction (Fig. 3c, d), highlighting a

common neural role underlying the onset of rosacea in disease families.

### *LRRC4/SH3PXD2A/SLC26A8* mutations promote the expression of vasoactive neuropeptides in human neural cells

Since the functional analyses of variant genes imply a link to neural function, we wondered whether the mutations in these genes would affect the development of rosacea via regulating neural function. We first detected the expression of LRRC4/SH3PXD2A/SLC26A8, three typical variant genes identified in discovery families and replicated in validation families, in different cell types of the human body via single-cell RNA-sequencing (scRNA-seq) analysis. Our results, as expected, showed that all three genes were highly expressed in the neural cells (Supplementary Fig. 3a), further suggesting a potential role of LRRC4/SH3PXD2A/SLC26A8 in regulating neural cell behaviors. We over-expressed mutant LRRC4/SH3PXD2A/SLC26A8 each with the corresponding identified variant in human neural cells (Supplementary Fig. 3b). According to the pathway analysis results (Fig. 3), we first examined the expression of cell adhesion-related molecules, such as

**Table 2 | Candidate genes validated in small rosacea families via WES**

| Family | Gene | Genomic position | Reference/alterative | cDNA change | Consequence | Amino acid change | GnomAD MAF | SIFT | Polyphen2 |
|---|---|---|---|---|---|---|---|---|---|
| 42 | LRRC43 | 12: 122684858 | T/C | c.T1472C | Missense | p.V491A | NA | 0.003, Deleterious | 0.915, Damaging |
| 332 | LRRC44 | 1: 74575239 | T/C | c.708-2 A > G | Splicing | NA | NA | NA | NA |
| 94 | LRRC47 | 1: 3700588 | T/G | c.A1282C | Missense | p.K428Q | NA | 0.005, Deleterious | 0.918, Damaging |
| 313 | LRRC55 | 11: 56949659 | C/T | c.C292T | Missense | p.R98W | NA | 0.017, Deleterious | 0.997, Damaging |
| 147 | LRRCC1 | 8: 56949659 | G/A | c.G1931A | Missense | p.R644H | 0.0008 | 0.000, Deleterious | 0.999, Damaging |
| 147 | LRRCC1 | 8: 86050476 | C/G | c.C2700G | Stopgain | p.Y900X | 0.0005 | NA | NA |
| 311 | LRRD1 | 7: 91788389 | G/C | c.C2046G | Stopgain | p.Y682X | 0.0004 | NA | NA |
| 314 | LRRD1 | 7: 91779953 | T/C | c.A2173G | Missense | p.I725V | 0.005 | 0.024, Deleterious | 0.984, Damaging |
| 399 | LRRD1 | 7: 91779953 | T/C | c.A2173G | Missense | p.I725V | 0.005 | 0.024, Deleterious | 0.984, Damaging |
| 48 | LRRTM4 | 2: 77746619 | T/C | c.A379G | Missense | p.N127D | 0.0015 | 0.004 Deleterious | 1.0, Damaging |
| 312 | SH3PXD2A | 10: 105363285 | C/T | c.G1606A | Missense | p.G536S | 0.0001 | 0.037, Deleterious | 1.0, Damaging |
| 319 | SLC26A8 | 6: 35945067 | A/G | c.T772C | Missense | p.S258P | 0.00001759 | 0.006 Deleterious | 0.999, Damaging |

GnomAD Genome Aggregation Database, MAF minor allele frequency, SIFT Sorting Intolerant From Tolerant, Polyphen2 Polymorphism Phenotyping v2, NA not applicable.

NCAM, ECAD, and CADM1, which play an essential role in regulating neural and synaptic function[21–23]. Our results demonstrated that NCAM, ECAD, and CADM1 were not affected in human neural cells harboring mutant *LRRC4/SH3PXD2A/SLC26A8* (Supplementary Fig. 3c–e). Neuropeptides are the key mediators, by which neurons regulate the behaviors of the local cells, including endothelial cells and immune cells, in the pathological and physiological processes of the skin[24]. Thus, we analyzed the expression levels of a series of neuropeptides, including PACAP, VIP, CGRPα, CGRPβ, NPY, TAC1, NGF, CALR, SST, and ADM2, and found that *PACAP* and *VIP* were increased in *LRRC4* mutant neural cells, *PACAP, NPY,* and *TAC1* increased in *SH3PXD2A* mutant cells, *PACAP, VIP, CGRPβ, CALR,* and *SST* increased in *SLC26A8* mutant cells (Fig. 4a–c and Supplementary Fig. 3f–h). By immunostaining analysis, we confirmed that PACAP was indeed upregulated at protein levels in human neural cells harboring mutant *LRRC4/SH3PXD2A/SLC26A8* (Fig. 4d). To further substantiate this finding, we co-immunostaining of PACAP and PGP9.5, a marker of intradermal nerve fibers[25] and demonstrated that PACAP was significantly increased in PGP9.5 positive neuron fibers in the lesional skin of rosacea patients (Fig. 4e, f). Collectively, our results demonstrate that mutation of *LRRC4/SH3PXD2A/SLC26A8* increases the production of vasoactive neuropeptides in human neural cells.

## L385P mutation in *Lrrc4* promotes rosacea development via peripheral neuron-derived neuropeptide VIP in mice

Considering that *LRRC4*, the only identified gene with a single rare deleterious variant in family 1 (Table 1), has been reported to play a critical role in regulating neural function, possibly by interacting with PAR complex, a key modulator of skin neurogenic inflammation[26–31], we focused on it to further investigate the role of variant genes in the pathogenesis of rosacea. We first established a knock-in mouse model harboring L385P mutation in *Lrrc4* gene, equivalent to the L386P mutation in human (Supplementary Fig. 4a, b). We then injected cathelicidin LL37 intradermally into wild-type (WT) mice and *Lrrc4* mutant mice, including heterozygotes (HET) and homozygotes (HOM), to induce rosacea-like mouse models as previously described[7,32], and compared the resulting rosacea-like phenotypes at different timepoints. We found that 12 h post the last LL37 administration, WT mice displayed obvious rosacea-like features, which were further exacerbated in mutant mice, and there was no significant difference between HET and HOM mice (Supplementary Fig. 4c, d). Our observation also showed that 36 h after the first LL37 injection, mutant mice had developed typical rosacea-like dermatitis, whereas WT mice did not exhibit apparent rosacea-like phenotypes until 12 h post the last LL37 administration (Supplementary Fig. 4e and Fig. 5a–c). Moreover, the average redness area and score at 36 and 48 h were remarkably increased in mutant mice (Fig. 5b, c). By immunohistochemistry (IHC) of CD31, we showed that the dilation of CD31⁺ blood vessels was further augmented in the lesional skin of mutant mice compared with WT mice after LL37 administration (Fig. 5d, e). Similarly, the dermis-infiltrating inflammatory cells and disease-characteristic inflammatory factors were also significantly increased in mutant mice (Fig. 5f, g and Supplementary Fig. 4f).

To further investigate the mechanism by which *Lrrc4* mutation facilitates rosacea development, we first examined the expression of *Lrrc4* in the indicated cell types, which may affect skin conditions in mice by using scRNA-seq data from the Tabula Muris Senis atlas[33]. Our results showed that *Lrrc4* is also highly expressed in neural cells compared with other cells in mice (Supplementary Fig. 5a), coincided with the results in humans (Supplementary Fig. 3a). We then performed RNA sequencing on the dorsal root ganglions (DRGs) from WT and mutant mice both injected with LL37. We identified 37 differentially expressed genes (DEGs) between mutant and WT DRGs ($P < 0.05$; Supplementary Data 3 and Fig. 6a), in which *Vip* was the top-ranked upregulated gene (Fig. 6b). By reverse transcription-quantitative

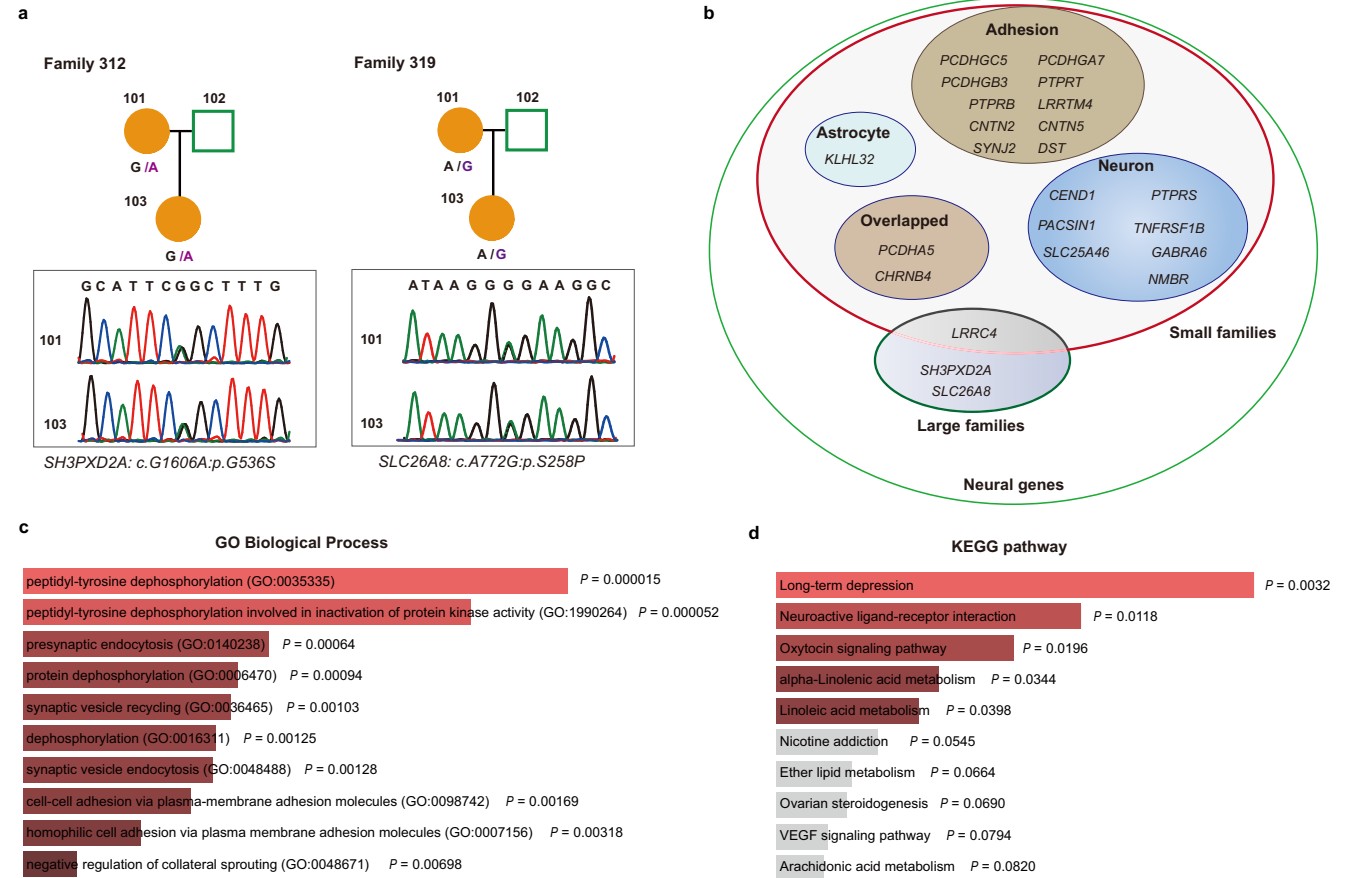

**Fig. 3 | Variant genes are replicated in small families and highlight the neural function. a** Additional variants in *SH3PXD2A* and *SLC26A8* were occurred in small families 312 and 319, respectively. **b** Functional category of neural-related gene set identified by WGS and WES, respectively, in large and small families. **c** Gene ontology (GO) analysis suggested neural synaptic processes and cell adhesion were functional categories for candidate genes. **d** KEGG pathway enrichment indicated long-term depression and neuroactive ligand-receptor interaction were significantly highlighted. The Fisher exact test (two-sided) was used for GO and KEGG enrichment.

polymerase chain reaction (RT-qPCR) and immunostaining, we confirmed that VIP was significantly increased in the DRG neurons of mutant compared with WT mice after LL37 administration, and other neuropeptides were not affected (Supplementary Fig. 5b and Fig. 6c, d). However, there existed no obvious changes in the expression of *Vip* and *Pacap* in the skin (Supplementary Fig. 5c). These results suggest a role of neuron-derived VIP in facilitating rosacea development in *Lrrc4* mutant mice.

By utilizing an antagonist peptide of VIP receptor, VIPhyb[34,35], we found that inhibition of VIP signaling could greatly improve rosacea-like dermatitis in *Lrrc4* mutant mice compared with the mice treated with scrambled VIPhyp peptides (sVIPhyp) (Fig. 6e, f). Considering VIP is a typical vasoactive neuropeptide, we detected the cutaneous vasodilation by IHC of CD31. Our results showed that blockade of VIP signaling by VIPhyb significantly declined the vasodilatation in the lesional skin of mutant mice administered with LL37 (Supplementary Fig. 5d and Fig. 6g). By histological and RT-qPCR analysis, we demonstrated that VIPhyb administration decreased the dermis-infiltrating cells and disease-characteristic inflammatory factors in mutant mice (Supplementary Fig. 5e and Fig. 6h, i). To exclude the possibility that VIPhyb injections might prevent local vasodilation and inflammatory cell infiltration caused by LL37 in skin. We injected LL37-treated WT mice with VIPhyb, and found that VIPhyb did not alleviate LL37-induced rosacea-like phenotypes in WT mice (Supplementary Fig. 6).

Taken together, these findings demonstrate that L385P mutation in *Lrrc4* promotes rosacea development, possibly by neuron-derived neuropeptide VIP in mice.

## Discussion

To date, there are few published studies designed to explore the genetic basis of rosacea, all of which utilized GWAS- and candidate gene-based methods to identify the common variants associated with rosacea in sporadic patients and healthy individuals[9,10,36]. However, common variants from these studies are difficult to be used to explain the susceptibility and familial aggregation of this disorder. In the present study, based on WGS of 3 large rosacea families (as discovery cohort) and WES of 49 additional small rosacea families (as validation cohort), we identified *LRRC4*, *SH3PXD2A*, and *SLC26A8*, each with a single rare genetic variant, as potential candidate genes for rosacea susceptibility. We further figured out the possible mechanisms by which mutations in these genes contribute to the development of rosacea via a series of experimentations in disease models. To the best of our knowledge, our study represents the first and currently the only WGS- and WES-based genetics study for rosacea, revealing the genetic basis of the pathogenesis and familial inheritance of this disorder.

The prevalence of rosacea is higher in fair-skinned people, especially in population with Celtic origin, whereas African Americans and Asians are less affected[1,37,38], suggesting a genetic link to the development of rosacea. However, there is a lack of genetic evidence. Until recently, several common variants were identified in *HLA-DRB1*, *HLA-DQA1*, *IRF4*, *SLC45A2*, and *IL13* genes in population of European descent[9,10,36,39]. However, these studies were all based on Caucasian population, and there still lack of identified genetic variants involved in the pathogenesis of rosacea in other ethnic and racial populations. In the present study, via WGS and WES in the Chinese population, we

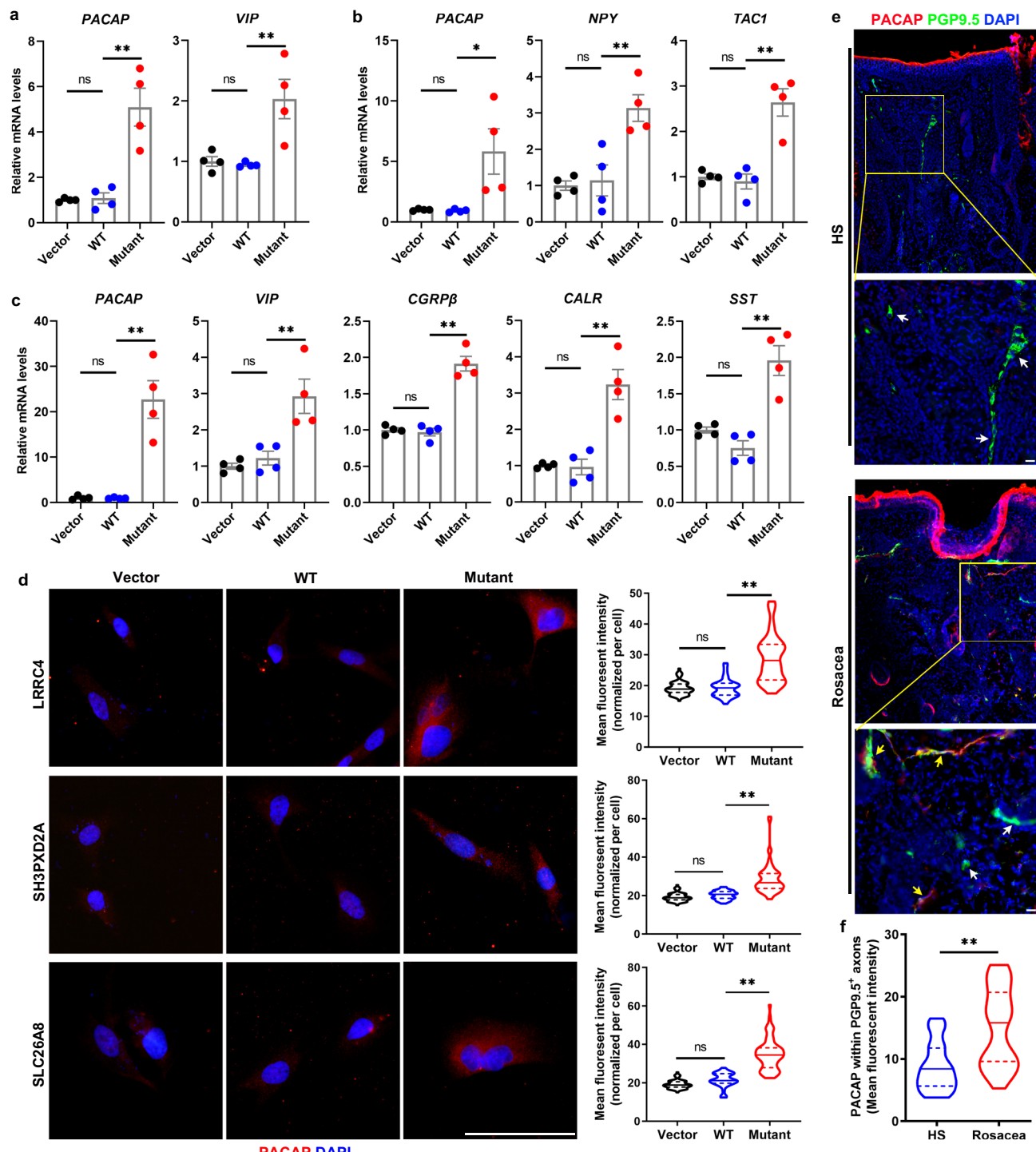

found a series of novel rare functional variants associated with rosacea, which are quite different from those previously identified in the Caucasian population.

Family history was up to 30% for rosacea[38,40], indicating a strong familial inheritance and offering us an excellent wealth to explore the susceptibility genes for this disease. Though there have been some studies revealing potential genetic elements of rosacea, they are all performed in sporadic patients[5], which are unlikely to explain the predisposition of familial aggregation. In this study, by employing a two-stage strategy, namely preliminary screening via WGS in large rosacea families and then validation by WES in additional families, we identified multiple rare genetic variants associated with the disease

susceptibility, and these variant genes from different families refer to a common characteristic of neural function. Pathway analyses of these genes further emphasize the functional roles in synaptic function, neural cell adhesion, long-term depression and neuroactive ligand-receptor interaction, consistent with our previous findings showing that the gene set involved in chemical synaptic transmission was highly upregulated in rosacea lesions[41]. Interestingly, long-term depression is the most prominently enriched pathway in KEGG analysis, providing a genetic explanation for the observations showing depression and anxiety are highly associated with rosacea in patients[42,43]. Collectively, these findings reveal genetic clues for the familial inheritance and a common role of neural function in the pathogenesis of rosacea.

**Fig. 4 | Mutation of *LRRC4/SH3PXD2A/SLC26A8* increases vasoactive neuropeptides in human neural cells. a–c** The relative mRNA expression levels of *PACAP, VIP, NPY, CGRPβ, TAC1, CALR*, and *SST* in human neural cells transfected respectively with *LRRC4* (**a**), *SH3PXD2A* (**b**), and *SLC26A8* (**c**) mutant/wild-type (WT)/Control vector plasmids (*n* = 4 biologically independent experiments). **a** *PACAP*: Mutant vs WT, *P* = 0.0009; WT vs Vector, *P* = 0.9992, *VIP*: Mutant vs WT, *P* = 0.0094; WT vs Vector, *P* = 0.9961. **b** *PACAP*: Mutant vs WT, *P* = 0.0324; WT vs Vector, *P* > 0.9999, *NPY*: Mutant vs WT, *P* = 0.0064; WT vs Vector, *P* = 0.9874, *TAC1*: Mutant vs WT, *P* = 0.0005; WT vs Vector, *P* = 0.9795. **c** *PACAP*: Mutant vs WT, *P* = 0.0004; WT vs Vector, *P* > 0.9999, *VIP*: Mutant vs WT, *P* = 0.0086; WT vs Vector, *P* = 0.9417, *CGRPβ*: Mutant vs WT, *P* < 0.0001; WT vs Vector, *P* = 0.9866, *CALR*: Mutant vs WT, *P* = 0.0006; WT vs Vector, *P* = 0.9996, *SST*: Mutant vs WT, *P* = 0.0004; WT vs Vector, *P* = 0.5298. **d** Immunostaining of *PACAP* in human neural cells transfected respectively with *LRRC4, SH3PXD2A,* and *SLC26A8* mutant/WT/control vector plasmids. Right panels, the quantification of mean fluorescent intensity for PACAP in the corresponding groups. *n* = 42–71 cells from three independent experiments. LRRC4: Mutant vs WT, *P* < 0.0001; WT vs Vector, *P* = 0.9976, SH3PXD2A: Mutant vs WT, *P* < 0.0001; WT vs Vector, *P* = 0.5517, SLC26A8: Mutant vs WT, *P* < 0.0001; WT vs Vector, *P* = 0.0612. **e** Co-immunostaining of PACAP and PGP9.5 on skin sections from rosacea patients (rosacea, *n* = 6) and healthy individuals (HS, *n* = 5). Higher-magnified images of yellow boxed areas are shown below the lower-magnified images for each group. Scale bar, 50 µm. Yellow arrowheads indicate PGP9.5 positive neuron axon with strong co-immunostaining signals of PACAP; White arrowheads indicate PGP9.5 positive neuron axon with low or no immunostaining signals of PACAP. **f** Quantification of mean fluorescent intensity for PACAP in PGP9.5 positive neuron fibers (*n* = 25 for HS; *n* = 35 for rosacea). *P* < 0.0001. DAPI staining (blue) indicates nuclear localization. Scale bar, 50 µm. All results are representative of at least three independent experiments. Data represent the mean ± SEM. *\*P* < 0.05, *\*\*P* < 0.01. ns indicates no significance. One-way ANOVA with Bonferroni's post hoc test (**a**–**d**) or two-tailed unpaired Student's *t* test (**f**) was used.

The identified variant genes associated with neural function include *LRRC4, SH3PXD2A,* and *SLC26A8*. They are all identified with a single rare deleterious variant in large rosacea families, and underscored in validation rosacea families. LRRC4, also known as Netrin-G ligand-2 (NGL2), is a member of the leucine rich repeat-containing (LRRC) family, and belongs to the superfamily of the *LRR* genes[26]. Previous studies have demonstrated that LRRC4 plays a critical role in the development of nervous system and neural function[26–29]. SH3PXD2A, namely TKS5, has been shown to be required for neural crest cell migration[44], and genetic evidence indicates the variants in SH3PXD2A are associated with neurofibromatosis[45]. SLC26A8 belongs to the SLC26 family, a conserved family of anion transporters, which interact with cystic fibrosis transmembrane conductance regulator (CFTR) to generate the SLC26-CFTR complex. Physical interaction between SLC26A8 and CFTR leads to CFTR stimulation[46], and CFTR is pivotal for the development and function of nervous system[47,48]. Consistently, we here demonstrate that these three genes are all highly expressed in neural cells, supporting their roles in regulating neural function. Besides, in three independent families we identified multiple variants in *PCDHA5*, a neural cadherin-like cell adhesion protein which has been considered to be essential for the establishment and function of specific cell–cell connections in the neural system[49,50]. Collectively, all these evidences further indicate the association of neural function with the pathogenesis of rosacea.

Although rosacea is well established as a chronic inflammatory skin disorder, flushing and burning sensations in the central face are also hallmark features of rosacea, which are considered to be primarily triggered by neurovascular dysregulation[1,5,51,52]. Recent studies have suggested that in rosacea lesions, neuronally expressed TRP channels, TLR2 and PAR₂ might respond to the triggers, resulting in the release of neurovascular and neuro-immune active neuropeptides, such as PACAP, VIP, substance P, CGRP and so on, and eventually lead to flushing, erythema and inflammation[5,6,20,53,54]. All these evidence sustain the hypothesis that rosacea is a kind of neurogenic skin inflammation. However, it remains unknown whether genetic components are involved in this aspect. Previous genetics studies have revealed some variants in several genes, which are mainly associated with immune responses in rosacea[5].

In this study, we identify multiple genes with rare variants associated with neural function. Among these genes, LRRC4 is highlighted, and has been previously reported to regulate neural cell behaviors via interacting with PAR complex, an important regulator of skin neurogenic inflammation[30,31]. Consistently, we here show that neural cells harboring the identified rare variant in *LRRC4* express more neurovascular and neuro-immune active neuropeptides, PACAP and VIP. Most importantly, our results demonstrated that in an LL37-induced rosacea mouse model, mutation in *Lrrc4* facilitates rosacea-like skin inflammation by neuropeptide VIP derived from peripheral neurons. Collectively, these findings might provide a genetic explanation for the hypothesis

that rosacea is a kind of neurogenic skin inflammation and establishes a model to study the neurogenic inflammation in rosacea. However, further study is needed to elucidate the precise mechanisms by which mutations of these genes affect the production of neuropeptides in neural cells; and it will be very interesting to determine how different variants regulate the same signaling axis mediated by neuropeptides.

Our study is not without limitations. It might be difficult to determine whether the alleviated rosacea-like symptoms in *Lrrc4* mutant mice are due to VIP inhibition or whether it is the combined PAC1/VPAC subtype receptor inhibition to play a role in improving symptoms, considering that VIP and PACAP both bind with high affinity to VPAC1, VPAC2 and PAC1 receptors, with the only difference that PACAP exhibits about 1000-fold higher affinity for PAC1 than VIP[55,56], and VIPhyb has also been reported to suppress peptide histidine isoleucine (PHI) and PACAP in addition to VIP[57,58]. Further experiments, for instance employing *Vip* conditional knockout mouse model, will be needed to clarify this issue.

In summary, our study reveals the substantial genetic heterogeneity, and the genetic basis of familial inheritance and neurogenic inflammation in the pathogenesis of rosacea, and suggests that the neural aspects should be considered in rosacea intervention.

## Methods

### Recruitment of rosacea families and disease diagnosis

This study was approved by the ethical committee of the Xiangya Hospital of Central South University. Three large rosacea families for WGS as our discovery cohort were obtained from the Han population of Hunan province of China (detailed information shown in Supplementary Data 4 and Fig. 2). In total, 49 small rosacea families for WES as our validation cohort were obtained from the Han population of Hunan province and its adjacent provinces of China (Shown in Supplementary Data 4 and Supplementary Fig. 2). The information on environmental factors and co-morbidities associated with rosacea has been collected and analyzed, provided in Supplementary Data 4. Written informed consents were acquired from every adult individuals and the parents/legal guardians of underage individuals. The diagnosis of rosacea in all families was ascertained as described below: (1) for individuals whose blood samples were collected, the diagnosis was performed in-person visits by 2 experienced dermatologists from the Department of Dermatology, Xiangya Hospital of Central South University, and the photos of the face were taken with informed consent in the sampling process; (2) for individuals whose blood samples were not collected, high-definition photos of face were obtained with informed consent, then the diagnosis was performed with these photos and combined with telephone consultations by three experienced dermatologists. Genomic DNA was extracted using HiPure Blood DNA Mini Kit (D3113, Magen Biotechnology, China). DNA Quality was evaluated by the following two means: (1) DNA degradation and contamination were determined on 1% agarose gels; (2) DNA

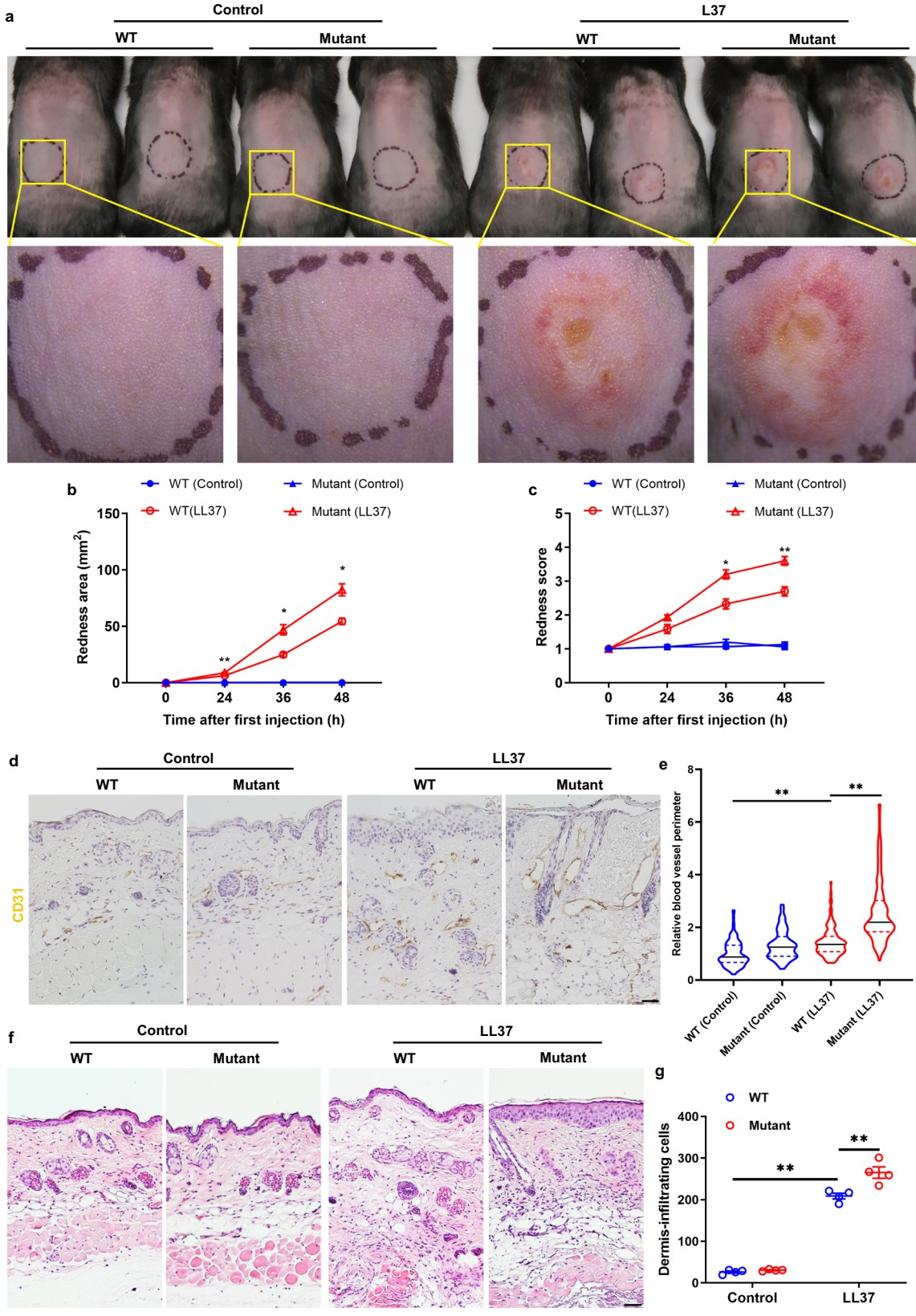

concentration was detected by the Qubit DNA Assay Kit and a Qubit 2.0 Fluorometer (ThermoFisher).

## WGS library generation

A total of 0.5 μg of genomic DNA for each sample from the participating individuals of large rosacea families, was utilized as input material for library generation. Libraries for sequencing were produced by the TruSeq Nano DNA HT Sample Prep Kit (Illumina) according to the manufacturer's instructions, and index codes were added to every sample. Briefly, via sonication, genomic DNA was fragmented to a median size of 350 bp. DNA fragments were then end-repaired, A-tailed, and ligated by using the full-length Illumina

**Fig. 5 | *Lrrc4* mutation facilitates the development of rosacea in mice. a** The back skins of WT and *Lrrc4* L385P mutant mice intradermally injected with LL37 or control vehicle. Images were taken 48 h after the first LL37 injection. Below panels, magnified images of yellow boxed areas. **b**, **c** The severity of the rosacea-like phenotypes after first LL37 injection for 24, 36, and 48 h, was assessed with the redness area (**b**) and score (**c**) (*n* = 5 for each group). *P < 0.01, **P < 0.01, comparison between *Lrrc4* mutant (LL37) and WT (LL37) group. **b** 24 h, *P* = 0.009; 36 h, *P* = 0.0197; 48 h, *P* = 0.0128. **c** 24 h, *P* = 0.1917; 36 h, *P* = 0.0109; 48 h, *P* = 0.0053. **d** Immunohistochemistry (IHC) of CD31 on skin sections from WT and *Lrrc4* mutant mice treated with LL37 or control vehicle. Scale bar, 50 μm. **e** Quantification of

relative blood vessel perimeter in the corresponding groups displayed with violin plot. *n* = 90–132 blood vessels from four independent mice for each group. Mutant (LL37) vs WT (LL37), *P* < 0.0001; WT (LL37) vs WT (Control), *P* < 0.0001. **f** HE staining of lesional skin sections from WT and mutant mice treated with LL37 or control vehicle. Scale bar, 50 μm. **g** Dermal infiltrating cells were quantified (*n* = 4 mice for each group). Mutant (LL37) vs WT (LL37), *P* = 0.0013; WT (LL37) vs WT (Control), *P* < 0.0001. All results are representative of at least three independent experiments. Data represent the mean ± SEM. *P < 0.05, **P < 0.01. Two-way ANOVA with Bonferroni's post hoc test was used.

sequencing adapters, and further PCR amplification was conducted. PCR products were purified with AMPure XP system. Sequencing libraries were analyzed for size distribution with an Agilent Bioanalyzer 2100 and were quantified by RT-qPCR.

## WES library generation
A total of 2 μg genomic DNA for each sample from the participating individuals of small rosacea families, was used to generate capture libraries with the Agilent SureSelect Human All Exon V6 kit (Agilent Technologies) according to the manufacturer's instructions. Fragmentation was conducted with a hydrodynamic shearing system (Covaris, Woburn, MA) to produce 180–280 bp fragments. DNA fragments were ligated with adapter molecules on both ends, and then selectively enriched via PCR followed by liquid-phase hybridization with biotin-labeled probes. A total of 60 Mb sequences of the whole human exome were acquired.

## Generation of sequencing data and quality control
WGS (19 patients/ healthy individuals from three large rosacea families) and WES (162 patients/healthy individuals from 49 small rosacea families) libraries were sequenced on the Illumina HiSeq X TEN platform (2 × 150-bp paired-end reads) (Novogene, Beijing, China). Read pairs were abandoned if: (a) either read included adaptor sequences (>10 nucleotides aligned to the adaptor, permitting ≤10% mismatches); (b) either read included more than 10% uncertain bases; or (c) either read included more than 50% low-quality bases (Phred quality <5). The following statistics were measured: total reads number, percentage of reads with average quality score >30, percentage of reads with average quality score >20, sequencing error rate, and GC content distribution.

## Sequencing data processing
Sequencing reads were mapped to the human reference genome (GRCh38) with BWA-MEM (v0.7.8). Unaligned reads that passed Illumina's quality filter (PF reads) were reserved. Picard tools were used to integrate data from multiple libraries and flow cell runs into a single BAM file for each sample. Only uniquely mapped, de-duplicated reads were retained for subsequent analyses. Quality scores were re-aligned with the Indel Realigner algorithm (GATK v3.8.0).

## Rare, pathogenic variant filtering
Rare variants with a minor allele frequency (MAF) of less than 1% were retained on the basis of dbSNP (v.137), 1000 Genomes Project data (August 2015, Chinese), Exome Aggregation Consortium (ExAC) and Novogene Bioinformatics Institute in-house exomeSeq databases, including 2573 exomes. Only nonsynonymous, frameshift, nonsense, and splice-site variants were selected. In silico functional analysis, SIFT, PolyPhen2, CADD, and GERP++ were used to predict the impact of each nonsynonymous variant on protein function. These different prediction software programs utilize algorithms to calculate the potential damage caused by a nucleotide variant by determining the likelihood of the substituted amino acid to affect protein function.

Family- and functional-based filtering was performed to identify potentially pathogenic variants according to the following criteria: (1)

variants were evaluated for co-segregation with rosacea phenotype based on an autosomal dominant inheritance; (2) the filtered-in variants was based on variant type (nonsynonymous, frameshift, nonsense, and splice site), minor allele frequency (MAF < 0.01); (3) functional predictions using the programs (SIFT score <0.05, PolyPhen2 score >0.825, CADD score >20 and GERP++ score >2, variants that were predicted to be damaging in at least three of the four algorithms).

## Functional enrichment and pathway analysis
To determine whether the candidate genes showed enrichment for specific biological pathways, Gene-set was input into EnrichR, a comprehensive gene set enrichment analysis web server 2016 update, (https://maayanlab.cloud/Enrichr/) for GO term enrichment analysis and KEGG pathway analysis[59]. Brain developmental gene-expression data were obtained from the human protein atlas.

## Human skin samples
All skin biopsies were acquired from the central face of female patients with rosacea or healthy individuals (aged 20–50 years) from the Department of Dermatology of Xiangya Hospital, Central South University. Rosacea patients were diagnosed with rosacea by clinical and pathologic examination. The utilization of human samples was approved by the ethical committee of the Xiangya Hospital of Central South University and written informed consent was acquired from all individuals, and the experiments were performed according to the principles set out in the WMA Declaration of Helsinki and the Department of Health and Human Services Belmont Report.

## Mice
Mice harboring L385P mutation in *Lrrc4* gene were generated by (Cyagen Biosciences, China). Heterozygote pairs were mated to generate wild-type, heterozygote, and homozygote mice for the subsequent experiments. For VIPhyb treatment, mice were intradermally injected with 10 μg VIPhyb (purity >95%, Sangon Biotech, China) or scrambled peptides as control peptide dissolved in 50 μl PBS per mouse daily as previously described[34,35]. All mice used in this study were sex-matched at 8 weeks, and kept in specific pathogen-free conditions with a regular 12 h light/12 dark cycle, at ~20–25 °C and 45–55% humidity. The experiments performed were according to the instructions and permissions of the ethical committee of the Xiangya Hospital of Central South University.

## LL37-induced rosacea-like mouse model
The LL37-induced rosacea-like mouse models were produced as previously described[7,32]. In brief, mice at 8-weeks-old were shaved one day before LL37 injection, then the indicated sites on the back skin were intradermally injected with 40 μl, 320 μM LL37 peptide (purity>95%, Sangon Biotech, China) or control vehicle twice a day for two days. Skin inflammation of the rosacea mouse model was assessed by the severity of erythema and edema as previously described[60]. Briefly, The redness score was assessed from 1 to 5, and 5 being the reddest. The redness area was determined by stereomicroscope measurements (Leica S8APO, Leica, Germany).

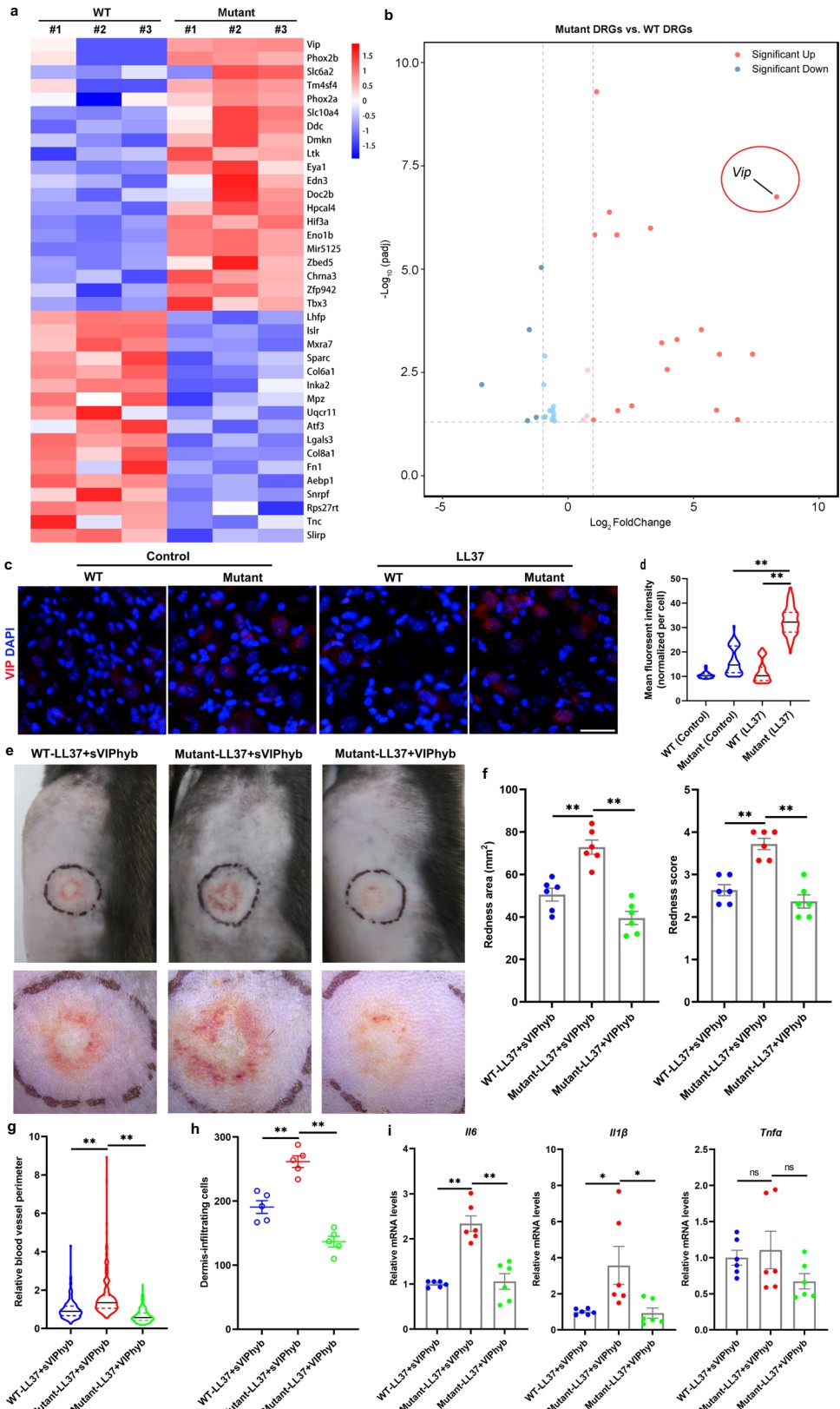

**RNA sequencing.** Total RNAs of mouse DRG samples were extracted with TRIzol Reagent (15596018CN, ThermoFisher). Library preparation and transcriptome sequencing were carried out using Illumina Nova-Seq 6000 (Novogene, Beijing, China). The mapping of 100-bp paired-end reads to genes was performed with HTSeq v0.6.0. The fragments per kilobase of transcript per million fragments mapped (FPKM) were analyzed, and differential expression analysis was conducted with the DESeq R package (1.10.1). The hierarchical clustering heatmap was generated with the ggplot library. Genes were considered significantly differently expressed at an adjusted $P$ value of <0.05.

**Fig. 6 | Blockade of VIP signaling alleviates rosacea-like phenotypes in mice harboring *Lrrc4* mutation. a** Heatmap of differentially regulated genes in DRGs from mutant and WT mice both injected with LL37 determined by RNA-sequencing (*n* = 3 independent biological samples for each group). Blue color denotes low FPKM expression; red, high FPKM expression. **b** Volcano plot of differentially regulated genes in DRGs from mutant and WT mice both injected with LL37 (*n* = 3 mice for each group). The red dots show the significantly upregulated genes; blue dots, significantly downregulated genes (*P* < 0.05). *Vip* is highlighted with red circle. **c** Immunostaining of VIP on sections of DRGs from WT and mutant mice treated with LL37 or control vehicle. Scale bar, 50 μm. **d** Quantification of mean fluorescent intensity for VIP in each neural cell in DRGs (*n* = 100 cells from three independent mice for each group). Mutant (LL37) vs WT (LL37), *P* < 0.0001; Mutant (LL37) vs Mutant (Control), *P* < 0.0001. **e** The back skins of LL37-administered WT and L385P mutant mice intradermally injected with VIPhyb or scrambled VIPhyp peptides (sVIPhyp). Images were taken 48 h after the first LL37 injection. Below panels,

magnified images of black dotted circle areas. **f** The severity of the rosacea-like features after first LL37 injection for 48 h, was evaluated with the redness area and score (*n* = 6 mice for each group). Redness area: Mutant-LL37+sVIPhyb vs WT-LL37+sVIPhyb, *P* = 0.0005; Mutant-LL37+VIPhyb vs Mutant-LL37+sVIPhyb, *P* < 0.0001. Redness score: Mutant-LL37+sVIPhyb vs WT-LL37+sVIPhyb, *P* = 0.0002; Mutant-LL37+VIPhyb vs Mutant-LL37+sVIPhyb, *P* < 0.0001. **g** Quantification of relative blood vessel perimeter in the corresponding groups presented with violin plot. **h** Dermal infiltrating cells were quantified (*n* = 5 mice for each group). Mutant-LL37+sVIPhyb vs WT-LL37+sVIPhyb, *P* = 0.0004; Mutant-LL37+VIPhyb vs Mutant-LL37+sVIPhyb, *P* < 0.0001. **i** The relative mRNA levels of *Il6, Il1β*, and *Tnfα* in lesional skins from LL37-treated WT and *Lrrc4* mutant mice intradermally injected with VIPhyb or sVIPhyp (*n* = 6 mice for each group). Data represent the mean ± SEM. **P* < 0.05, ***P* < 0.01. ns indicates no significance. One-way ANOVA with Bonferroni's post hoc test was used.

**Gene-expression analyses with scRNA-seq data.** The scRNA-seq expression datasets of different cell types in human were downloaded from the Human Protein Atlas database (https://www.proteinatlas.org). The scRNA-seq data from the Tabula Muris Senis atlas were acquired from the Gene Expression Omnibus (GSE149590)[33]; briefly, single-cell raw counts and cell annotations were downloaded from the file of GSM4505404, and the raw data were normalized with the Seurat R package[61]. Gene-expression data have been extracted from cells which may affect skin conditions (Endothelial cells, B cells, macrophages, skin epidermal cells, T cells, neural cells, fibroblasts, dendritic cells, NK cells, smooth muscle cells, and neutrophils) in young mice (3 months).

### Histological analysis
The histological analysis was performed as previously described[62,63]. Briefly, skin samples were fixed in formalin, and then embedded in paraffin. Sections were stained with hematoxylin and eosin (HE). The infiltrating cell number in the dermis was regarded as histological features. For calculating infiltrating cells in the dermis, five areas (0.444 square inch for one) on each section were randomly chosen, and the infiltrating cell number in the dermis was calculated.

### RT-qPCR
Total RNA was extracted from mice skin tissues, DRGs, and human neural cells with TRIzol Reagent, and the NanoDrop spectrophotometer (ThermoFisher) was used for RNA quality assessment. mRNA was reverse-transcribed via using the Maxima H Minus First Strand cDNA Synthesis Kit with dsDNase (K1682, ThermoFisher) according to the manufacturer's instructions. qPCR was performed by using iTaq™ Universal SYBR® Green Supermix (Bio-Rad) on a Light-Cycler 96 (Roche) thermocycler. The relative expression for each gene was calculated via the delta CT method relative to internal control gene GAPDH. The fold change for each gene was normalized to the control group. The sequences of specific primers for each gene are listed in Supplementary Data 5.

### Immunofluorescence
Immunofluorescence for mice skin and DRG sections, and cultured human neural cells was performed as previously described[7]. In brief, frozen sections (10 μm) and neural cells plated on glass coverslips in 24-well plates were fixed for 10 min with 4% PFA, followed by PBS washed three times, and blocked for 60 min with blocking buffer (5% NDS, 1% BSA, 0.3% Triton X-100). The indicated primary antibodies were incubated overnight at 4 °C. Alexa Fluor 594-conjugated secondary antibody (1:1000, A-21207, ThermoFisher) was incubated for 60 min at room temperature. After washing with PBS, sections were counterstained with DAPI. All images were taken with a Zeiss Axioplan 2 microscope. The quantification of fluorescence intensity corresponding

to the sum of the gray values of all the pixels in each cell divided by the pixel number per cell and the measurement of cell surface were conducted with ImageJ (this analysis normalizes the number of pixels with the cell size). The following primary antibodies were used: Rabbit anti-PACAP (1:200, ab181205, Abcam), Rabbit anti-VIP (1:200, ab272726, Abcam).

### Immunohistochemistry
Human and mouse skin samples were fixed in formalin and embedded in paraffin, and skin sections were cut and used. Immunohistochemistry for mouse skin sections (5 μm) was performed as previously described[64]. As negative controls, the primary antibodies were omitted. Images were taken from three typical areas for each sample. To evaluate the dilation of cutaneous blood vessels, the perimeter of CD31-positive blood vessels were calculated by ImageJ. Double immunohistochemistry for human skin sections (30 μm) was conducted by using the Opal 4 color manual immunohistochemistry (IHC) kit (NEL810001KT, PerkinElmer). Images were taken with a Zeiss Axioplan 2 microscope. The quantification of fluorescence intensity corresponding to the sum of the gray values of all the pixels in each cell divided by the pixel number per cell and the measurement of cell surface were conducted with ImageJ (this analysis normalizes the number of pixels with the cell size). The following primary antibodies were used: Rabbit anti-CD31 (1:100, 77699, Cell Signaling), Rabbit anti-PACAP (1:200, ab181205, Abcam), Rabbit anti-PGP9.5 (1:200, ab108986, Abcam).

### Plasmids and cell culture
The sequences of WT and mutant *LRRC4, SH3PXD2A,* and *SLC26A8* each harboring the indicated variant identified in large rosacea families were cloned into vector pLVX-IRES-Puro (Addgene). Primary human neural cell line HCN-2 was obtained from ATCC (CRL-10742) at passage 13, and the cells were expanded and used at passage 16–17 in this study. Cells were cultured in DMEM supplemented with 10% fetal bovine serum, 4 mM glutamine, penicillin-streptomycin (ThermoFisher). Primary human neural cells were transfected with WT or mutant *LRRC4/SH3PXD2A/SLC26A8* vectors or control vectors by using Transfection reagent FugeneHD (E2311, Promega). Forty-eight hours after transfection, cells were harvested for total RNA extraction. For immunofluorescence, human neural cells plated on glass coverslips in 24-well plates were transfected, and 72 h after transfection cells were harvested. All experiments were performed at least three times.

### Statistical analysis
Statistical analysis was performed with GraphPad 8.0. Data are presented as the mean ± SEM. We determined the data for normal distribution and similar variance between groups. One-way analysis of variance (ANOVA) with relevant post hoc tests for multiple

comparisons or two-tailed unpaired Student's $t$ test for comparisons between two groups was used to calculate statistical significance ($*P < 0.05$, $**P < 0.01$). When the data were not normally distributed or displayed unequal variances between groups, we conducted statistical analysis by two-tailed Mann–Whitney $U$ test. No statistical method was used to predetermine the size of the samples. Animals in the present study were randomly allocated to different groups.

### Reporting summary

Further information on research design is available in the Nature Portfolio Reporting Summary linked to this article.

## Data availability

All data needed to assess the conclusions in the present study are provided in the manuscript and/or the Supplementary Materials. The uploading and sharing of the genetic data of participated individuals in this project is not permissible in terms of a review by the Human Genetic Resources Administration of China based on regulations documented in the Interim Measures for the Administration of Human Genetic Resources. We have listed the summaries of the mutation data as detailed as possible, and these details are available to other researchers, including the exonic and splicing mutations in large and small rosacea families (shown in Supplementary Data 1 and Supplementary Data 2, respectively). scRNA-seq datasets of different cell types in the human body were downloaded from the Human Protein Atlas database (https://www.proteinatlas.org). scRNA-seq data from the Tabula Muris Senis atlas were obtained from the Gene-Expression Omnibus (accession number: GSE149590). The codes of our study were present in GitHub (https://github.com/nanlandetian/SkinAging). Sequencing data from DRGs of *Lrrc4* mutant and WT mice have been deposited in the genome sequence archive under accession number CRA009850 (https://ngdc.cncb.ac.cn/gsa/). The Human reference (GRCh38) dataset required for analysis is available at https://www.ncbi.nlm.nih.gov/assembly/GCF_000001405.26/. Other data supporting the findings of the present study are required to contact with the corresponding author (Ji Li, E-mail: liji_xy@csu.edu.cn) for identity verification purposes under adhering to the Chinese regulations. Source data are provided with this paper.

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

## Acknowledgements

This work was supported by the National Key Research and Develop-ment Program of China (No. 2021YFF1201205), the National Natural Science Funds for Distinguished Young Scholars (No. 82225039), the National Natural Science Foundation of China (Nos. 81874251, 82073457, 82003385, and 82173448), the Natural Science Foundation of Hunan Province, China (No. 2020JJ5888), and by the Science and Technology Innovation Plan of Hunan province (No. 2018SK2087). We thank our colleagues (Department of Dermatology, Xiangya Hospital, Central South University, China) for their generous support throughout this work.

## Author contributions

J.L. and Z.D. designed and conceived the study. G.Z., Z.D., and M.C. performed data analyses. G.Z. and Z.D. performed basic WGS and WES analysis. Z.Z., W.X., T.L., W.S., D.J., B.W., F.L., Y.T., Y.H., Y.Z., and Q.W. contributed to sample collection. Q.P. helped to generate sequencing libraries. Z.W. helped to perform Sanger sequencing validation. M.C. and Z.D. performed cell and staining experiments. M.C. and Z.D. performed mouse experiments. S.X. assisted with molecular cloning. L.S. and H.X. provided critical discussion and suggestions. Z.D., G.Z., J.L., and M.C. prepared the manuscript with input from coauthors.

## Competing interests

The authors declare no competing interests.
