## [Peer Review File · Nature Communications]

Whole genome sequencing identifies genetic variants associated with neurogenic inflammation in rosaceaREVIEWER COMMENTS

Reviewer #1 (Remarks to the Author):

The present study from Zhili Deng and collaborators is on Rosacea pathogenesis, specifically its genetic component. The authors analyze three families affected by rosacea for 13 patients and six controls, plus what they define as small families (49 families with 114 affected members and 48 nonaffected members). The study has been conducted using GWAS and selecting genes that have a function in the neural system.

The authors found a cluster of genes that influence the expression of neuropeptides.

The most critical mutations were reproduced in cells and in mice to show a higher propensity to respond to Cathelicidin as a model of rosacea-like inflammation.

The study is innovative. It reinforces previous knowledge that a neuropeptide imbalance is a significant cause of rosacea. The data presented are well-elaborated and well-presented and support the paper's claims.

It is vital to skin biology and the dermatology world.

Major comments:

-

- The abstract needs English revisions.
- The authors should be more careful with abbreviations and full names

Reviewer #2 (Remarks to the Author):

In this paper, the authors identify novel genetic variants associated with the familial form of rosacea. However, the authors could only identify deleterious variants in the LRR4, SH3PXD2A and SLC26A8 genes, the performed functional investigations are very convincing. The authors proved that the identified variants increase the production of neuropeptides involved in the pathogenesis of rosacea.

Variants of the LRR4, SH3PXD2A and SLC26A8 genes have not been associated with rosacea previously. Therefore, these results are great novelties in the elucidation of the genetic background of rosacea.

Questions:

1. Is the presence of the identified variants enough to initiate rosacea? Or they only increase the susceptibility to develop rosacea?
2. Genetic contribution to rosacea is app. 46%, while environmental factors contribute to 54%. Did the authors collected information about environmental factors contributing to disease development among the investigated patients?
3. What was the frequency of the co-morbidities of rosacea? Did the patients with LRR4, SH3PXD2A and SLC26A8 variants suffered any of them?
4. Previous investigations identified endoplasmic reticulum stress secondary to environmental triggers, production of antimicrobial peptides, neurogenic inflammation, and downstream activation of other innate and adaptive immunity inflammatory mediators as key factors in the mechanism of rosacea. How the encoded proteins of the identified novel candidate genes contribute to this big picture?

Reviewer #3 (Remarks to the Author):

The authors have identified genes associated with rosacea through whole genome and whole exome sequencing and claim to have validated these findings from other families. However, several important issues remain to be clarified:

- 1) please clarify the ethnicity of the initial and validation families, as not all family members may be Han Chinese. Lack of uniform genetic background could lead to false positives
- 2) how were the rosacea diagnoses ascertained? was any dermatologist involved? was this based on photos, in person visits or both? please specify
- 3) is there a link where we can access the actual raw data to confirm the results? We would need to see where the validation families come from? all Han Chinese or no?
- 4) how do we know if there were any co-morbidities that these human subjects may have had to confound the findings for instance, alcohol intolerance, concurrent neurological condition, etc.

Reviewer #4 (Remarks to the Author):

DISCLOSURE: Comments on the manuscript (NCOMMS-22-45806) reported by this reviewer relate only to the validity of VIP experiments. As such, results from deep whole genome sequencing (WGS) and validation studies using whole exome sequencing (WES) will not be assessed.

Dear Editor,

The manuscript "Whole genome sequencing reveals genetic variants in genes 2 associated with neurogenic inflammation in rosacea" submitted by Dr Zhili Deng and collaborators aims to identify rare and potentially deleterious genetic variants using WGS (experimental cohort) and WES (validation cohort).

The study identifies single mutations in *LRRC4*, *SH3PXD2A* and *SLC26A8* genes. They also find splice site and missense variants in LRR family genes in the validation cohort. Additional functional studies using gene ontology and KEGG databases revealed clustering of genes associated with neuronal function. Therefore, authors perform mechanistic studies to determine if mutations in the identified genes could provide an explanation for the neurogenic relationship with rosacea. Single-cell RNA sequencing confirmed high levels of expression *LRRC4*, *SH3PXD2A* and *SLC26A8* genes in neuronal cells. Overexpression of the identified mutant genes in human neural cells (HCN-2 cell line) demonstrated how mutations of *LRRC4*, *SH3PXD2A* and *SLC26A8* individually contributed to increase transcript levels of several neuropeptides, including PACAP and VIP, further confirmed by immunocytochemistry (in neurons) and co-immunofluorescence with the PGP9.5 marker in rosacea skin samples. Finally, to characterise *LRRC4*, the only identified gene with a single rare deleterious variant in family 1, authors utilise a knock-in mouse harbouring an equivalent gene mutation to that identified in humans and inject these mice intradermally with cathelicidin (LL37) and monitor rosacea-like skin manifestations in wild-type, heterozygous and homozygous mutant animals. Here, they show that mutant mice exhibit exacerbated rosacea-like skin manifestations than wild-types. In dorsal root ganglia, mutant mice display increased expression of VIP. Treatment of these mice with VIP hybrid (VIPHyb), a putative VIP antagonist, partly reversed the redness and local inflammatory response in LL37-injected mice.

Altogether, the work is coherent with the hypothesis that the identified rare genetic mutations can indeed contribute to the vulnerability to rosacea and provide a convincing link with the neurogenic nature of the disease. This reviewer also appreciated the design of the numerous mechanistic experiments to test the effects of these mutations on PACAP and VIP expression in neurons and peripheral nerve terminals.

There are, however, a few reservations related to the use of VIPHyb as a VIP antagonist, mainly due to its lack of receptor specificity. Details are provided below:

VIP and PACAP both bind with high affinity to VPAC1, VPAC2 and PAC1 receptors, with the only distinction that PACAP displays about 1000-fold higher affinity for PAC1 than VIP (PMID: 17574305, PMID: 35897648). VIPHyb has been reported to also inhibit PHI and PACAP, in addition to VIP (PMID: 11859929, PMID: 14706566). Therefore, it is difficult to determine whether the reduced skin

manifestations in mutant mice are due to VIP inhibition or whether it is the combined PAC1/VPAC subtype receptor inhibition to play a role in ameliorating symptoms. This reviewer is aware that specific VIP receptor antagonists are lacking and, although this aspect is not critical, additional experiments in animals also carrying a VIP null-mutation would be beneficial. Alternatively, the authors are recommended to highlight this limitation of the study.

Another important aspect is that VIP, as well as PACAP, are notorious for their vasodilator properties (PMID: 15959462, PMID: 15155712, PMID: 24563332), which may be prevented by administration of available VIP antagonists (PMID: 2116730). As such, it cannot be excluded that VIPHyb injections prevent local vasodilation and immune cells infiltration caused by LL37 in mutant mice. These considerations are important to determine a causal link with VIP induction in these animals.

Other less important remarks relate to the use of PGP9.5 as a neural marker. PGP9.5 acts as a tissue-specific ubiquitin carboxyl terminal hydrolase isoenzyme, and is also known as UCH-L1. It is abundant in peripheral nerves, and broadly used to detect intra-epidermal nerve fibres (PMID: 33892490, PMID: 35295465, PMID: 28914336). Whilst this makes PGP9.5 a suitable marker to co-localise nerve terminals in the skin, they should cautiously refer to it as a marker of intra-dermal nerve fibres rather than neural marker.

Finally, HCN-2 cortical neuron cell line (ATCC, CRL-10742) undergo senescence relatively quickly (> passage 21), so it is important that the authors indicate the passage number in the methods.

Reviewer #1

Overall comments:

The present study from Zhili Deng and collaborators is on Rosacea pathogenesis, specifically its genetic component. The authors analyze three families affected by rosacea for 13 patients and six controls, plus what they define as small families (49 families with 114 affected members and 48 nonaffected members). The study has been conducted using GWAS and selecting genes that have a function in the neural system. The authors found a cluster of genes that influence the expression of neuropeptides. The most critical mutations were reproduced in cells and in mice to show a higher propensity to respond to Cathelicidin as a model of rosacea-like inflammation.

The study is innovative. It reinforces previous knowledge that a neuropeptide imbalance is a significant cause of rosacea. The data presented are well-elaborated and well-presented and support the paper's claims. It is vital to skin biology and the dermatology world.

Overall response:

We thank the reviewer for the recognition of our work and the helpful suggestions, all of which have been addressed, as detailed below.

Major comment 1:

The abstract needs English revisions.

Response 1:

We thank the reviewer for the kind suggestion. As suggested, we have revised the abstract carefully.

Major comment 2:

The authors should be more careful with abbreviations and full names.

Response 2:

We thank the reviewer for the kind suggestion. As suggested, we have carefully corrected the abbreviations and full names throughout the manuscript.

Reviewer #2

Overall comments:

In this paper, the authors identify novel genetic variants associated with the familial form of rosacea. However, the authors could only identify deleterious variants in the LRR4, SH3PXD2A and SLC26A8 genes, the performed functional investigations are very convincing. The authors proved that the identified variants increase the production of neuropeptides involved in the pathogenesis of rosacea.

Variants of the LRR4, SH3PXD2A and SLC26A8 genes have not been associated with rosacea previously. Therefore, these results are great novelties in the elucidation of the genetic background of rosacea.

Overall response:

We thank the reviewer for the recognition of our work and the insightful comments, all of which have been addressed, as detailed below.

Comment 1:

1. Is the presence of the identified variants enough to initiate rosacea? Or they only increase the susceptibility to develop rosacea?

Response 1:

We thank the reviewer for the insightful comments. It is a very interesting question. Existing evidences indicate that rosacea is a multifactorial disease, in which genetic factors, dysregulation of the innate and adaptive immune system, vascular and neuronal dysfunction, and microorganisms appear to be involved. Triggers such as heat, stress, ultraviolet light, spicy food, hot drinks, smoking, and alcohol may exacerbate symptoms. Here, to the best of our knowledge, we proposed that the identified variants, referred to the genetic background, play roles in increasing the susceptibility of rosacea or aggravating the development of rosacea under the same challenging conditions (including environmental

triggers, dysregulated immune and neurovascular systems), considering but not limited to the following reason: mice harboring the variant in *Lrrc4* gene, equivalent to the identified variant in humans, do not develop rosacea spontaneously, but are more sensitive to the challenging condition, such as LL37 injection, which has been reported to induce rosacea-like phenotypes.

Comment 2:

2. Genetic contribution to rosacea is app. 46%, while environmental factors contribute to 54%. Did the authors collect information about environmental factors contributing to disease development among the investigated patients?

Response 2:

We thank the reviewer for the insightful and helpful comments. We agreed that the pathogenesis of rosacea is involved in both genetic contribution and environmental factors. In fact, we had collected the information about environmental factors contributing to disease development among the investigated subjects before. In the revised manuscript, we have provided the detailed information and performed further analysis with these data, and found that in addition to the shared variants, the environmental factors may also contribute to rosacea development among the investigated subjects, such as heat (OR: 42.992; 95%CI: 10.579-174.718), UV exposure (OR: 6.837; 95%CI: 1.195-39.112) (**shown in Supplementary Data 4**). And we have also described these data in the Methods of revised manuscript.

Comment 3:

3. What was the frequency of the co-morbidities of rosacea? Did the patients with *LRRC4*, *SH3PXD2A* and *SLC26A8* variants suffer any of them?

Response 3:

We thank the reviewer for the helpful comments. As suggested, we have provided the detailed information of the patients, including co-morbidities, and performed further analysis with these data, and found that hypertension (Frequency: 11.6788%) and diabetes (Frequency: 4.3795%) are the main co-morbidities among the investigated patients, and certain patients with *LRRC4*,

SH3PXD2A and SLC26A8 variants suffered hypertension (**shown in Supplementary Data 4**).

Comment 4:

4. Previous investigations identified endoplasmic reticulum stress secondary to environmental triggers, production of antimicrobial peptides, neurogenic inflammation, and downstream activation of other innate and adaptive immunity inflammatory mediators as key factors in the mechanism of rosacea. How the encoded proteins of the identified novel candidate genes contribute to this big picture?

Response 4:

We thank the reviewer for the recognition of our work. In the present study, we have demonstrated that the identified novel variant genes (including *LRRC4*, *SH3PXD2A* and *SLC26A8*) are highly expressed in the neural cells, and mutations of these genes induce the production of neuropeptides, which are responsible for the development of rosacea. Particularly, *Lrrc4* mutation promotes rosacea-like skin inflammation via neuropeptide VIP derived from peripheral neurons in mice. Therefore, we proposed that the regulatory network mediated by the encoded proteins of the identified novel candidate genes is very likely to be an important part of neurogenic inflammation in this big picture of rosacea development. And we have added this issue in the discussion part of the manuscript.

Reviewer #3

Overall comments:

The authors have identified genes associated with rosacea through whole genome and whole exome sequencing and claim to have validated these findings from other families. However, several important issues remain to be clarified:

Overall response:

We thank the reviewer for the recognition of our work, and the insightful and helpful comments, all of which have been addressed, as detailed below.

Comment 1:

1) please clarify the ethnicity of the initial and validation families, as not all family members may be Han Chinese. Lack of uniform genetic background could lead to false positives.

Response 1:

We thank the reviewer for the kind suggestion. At the beginning of the study, we had payed attention to the ethnicity of rosacea families. We selectively collected the rosacea families all from Hunan province and its adjacent provinces of China, in which all members are Han Chinese. Therefore, all families used in the present study have uniform genetic background, and all the detailed information on this issue has been provided in **the revised Methods** and **Supplementary Data 4**.

Comment 2:

2) how were the rosacea diagnoses ascertained? was any dermatologist involved? was this based on photos, in person visits or both? please specify.

Response 2:

We thank the reviewer for the kind suggestion. In the present study, the diagnosis of rosacea in all families was ascertained as described below: 1) for individuals whose blood samples were collected, the diagnosis was performed in person visits by 2 experienced dermatologists, and the photos of face were taken with informed consent in the sampling process; 2) for individuals whose blood samples were not collected, high-definition photos of face were obtained with informed consent, then the diagnosis was performed with these photos and combined with telephone consultations by 3 experienced dermatologists. And we have added this information to the Methods part of the revised manuscript.

Comment 3:

3) is there a link where we can access the actual raw data to confirm the results?
We would need to see where the validation families come from? all Han Chinese or no?

Response 3:

We thank the reviewer for the interest in our work.

1. For the question “is there a link where we can access the actual raw data to confirm the results?”

Response: Although the uploading and sharing of the genetic rawdata of participated subjects, generated from WGS or WES, is not permissible in terms of a review by the Human Genetic Resources Administration of China based on regulations documented in the Interim Measures for the Administration of Human Genetic Resources, we have listed the summaries of all the mutation data as detailed as possible, and these details are available to other researchers, including the exonic and splicing mutations in large and validation rosacea families (**shown in Supplementary Data 1 and Supplementary Data 2**, respectively). If other data supporting the findings of the present study are required, welcome to contact with the corresponding author (Ji Li, Email: liji_xy@csu.edu.cn) for identity verification purposes under adhering to the Chinese regulations after the manuscript is published.

2. For the question “We would need to see where the validation families come from? all Han Chinese or no?”

Response: All the validation families are from Hunan province and its adjacent provinces of China, in which all members are Han Chinese, and we have provided all the detailed information on this issue in **the revised Methods** and **Supplementary Data 4**.

Comment 4:

4) how do we know if there were any co-morbidities that these human subjects may have had to confound the findings for instance, alcohol intolerance, concurrent neurological condition, etc.

Response 4:

We thank the reviewer for the insightful and helpful comments. We had collected the information about the environmental triggers, lifestyles and co-morbidities (including alcohol intolerance, concurrent neurological condition, etc.) among the investigated subjects before. In the revised manuscript, we have provided the detailed information and performed further analysis with these data, and found that among the investigated patients in this study, hypertension (Frequency: 11.6788%) and diabetes (Frequency: 4.3795%) are the main co-morbidities, which was consistent with the previous study (J Am Acad Dermatol. 2018 Jan;78(1):167-170); the frequency of alcohol intolerance is 2.92%, but concurrent neurological condition (including Parkinson's disease and Alzheimer's disease) was not found (**shown in Supplementary Data 4**). And we have also described these information in the Methods of revised manuscript.

Reviewer #4

Overall comments:

The manuscript "Whole genome sequencing reveals genetic variants in genes associated with neurogenic inflammation in rosacea" submitted by Dr Zhili Deng and collaborators aims to identify rare and potentially deleterious genetic variants using WGS (experimental cohort) and WES (validation cohort). The study identifies single mutations in LRRC4, SH3PXD2A and SLC26A8 genes. They also find splice site and missense variants in LRR family genes in the validation cohort. Additional functional studies using gene ontology and KEGG databases revealed clustering of genes associated with neuronal function. Therefore, authors perform mechanistic studies to determine if mutations in the identified genes could provide an explanation for the neurogenic relationship with rosacea. Single-cell RNA sequencing confirmed high levels of expression LRRC4, SH3PXD2A and SLC26A8 genes in neuronal cells. Overexpression of the identified mutant genes in human neural cells (HCN-2 cell line)

demonstrated how mutations of LRRC4, SH3PXD2A and SLC26A8 individually contributed to increase transcript levels of several neuropeptides, including PACAP and VIP, further confirmed by immunocytochemistry (in neurons) and co-immunofluorescence with the PGP9.5 marker in rosacea skin samples. Finally, to characterise LRRC4, the only identified gene with a single rare deleterious variant in family 1, authors utilise a knock-in mouse harbouring an equivalent gene mutation to that identified in humans and inject these mice intradermally with cathelicidin (LL37) and monitor rosacea-like skin manifestations in wild-type, heterozygous and homozygous mutant animals. Here, they show that mutant mice exhibit exacerbated rosacea-like skin manifestations than wild-types. In dorsal root ganglia, mutant mice display increased expression of VIP. Treatment of these mice with VIP hybrid (VIPHyb), a putative VIP antagonist, partly reversed the redness and local inflammatory response in LL37-injected mice.

Altogether, the work is coherent with the hypothesis that the identified rare genetic mutations can indeed contribute to the vulnerability to rosacea and provide a convincing link with the neurogenic nature of the disease. This reviewer also appreciated the design of the numerous mechanistic experiments to test the effects of these mutations on PACAP and VIP expression in neurons and peripheral nerve terminals.

There are, however, a few reservations related to the use of VIPHyb as a VIP antagonist, mainly due to its lack of receptor specificity. Details are provided below:

Overall response:

We thank the reviewer for the recognition of our work, and the insightful and helpful comments, all of which have been addressed, as detailed below.

Comment 1:

VIP and PACAP both bind with high affinity to VPAC1, VPAC2 and PAC1 receptors, with the only distinction that PACAP displays about 1000-fold higher

affinity for PAC1 than VIP (PMID: 17574305, PMID: 35897648). VIPHyb has been reported to also inhibit PHI and PACAP, in addition to VIP (PMID: 11859929, PMID: 14706566). Therefore, it is difficult to determine whether the reduced skin manifestations in mutant mice are due to VIP inhibition or whether it is the combined PAC1/VPAC subtype receptor inhibition to play a role in ameliorating symptoms. This reviewer is aware that specific VIP receptor antagonists are lacking and, although this aspect is not critical, additional experiments in animals also carrying a VIP null-mutation would be beneficial. Alternatively, the authors are recommended to highlight this limitation of the study.

Response 1:

We thank the reviewer for the helpful and constructive comments. We agreed that it might be difficult to determine whether the reduced skin manifestations in mutant mice are due to VIP inhibition or whether it is the combined PAC1/VPAC subtype receptor inhibition to play a role in ameliorating symptoms, considering that VIP and PACAP both bind with high affinity to VPAC1, VPAC2 and PAC1 receptors, with the only distinction that PACAP displays about 1000-fold higher affinity for PAC1 than VIP, and VIPHyb has been reported to also inhibit PHI and PACAP in addition to VIP.

In the present manuscript, we showed that in rosacea-like mouse models, only VIP was increased, while PACAP and other neuropeptides were not affected in the DRG neurons of mutant mice (**shown in revised Supplementary Figure 5b, Figure 6c and d**); VIPHyb injections could alleviate the exacerbation of rosacea-like symptoms induced by *Lrrc4* mutation (**shown in revised Figure 6 and Supplementary Figure 5**). In revised manuscript, to determine whether the mutation would affect the neuropeptides (including VIP and PACAP) in skin, we detected their expression and found that the expression of VIP and PACAP was not affected in the skin lesions of mutant mice (**shown in new Supplementary Figure 5c**); we also showed that *Lrrc4* is mainly expressed in the neural cells in mice via single-cell RNA sequencing

analysis (**shown in new Supplementary Figure 5a**), which is consistent with the results in humans (shown in Supplementary Figure 3a); moreover, we performed bulk RNA-sequencing, and found that *Vip* is indeed the only neuropeptide significantly increased in the DRGs of mutant mice after LL37 administration, while other neuropeptides (including *Pacap*) are not affected (**shown in new Figure 6a and b, and Supplementary Data 3**). All our additional data provided in the revised manuscript at least indirectly support our conclusion that *Lrrc4* mutation promotes rosacea development via neuropeptide VIP derived from peripheral neurons. We have also, as suggested, highlighted this limitation of the study in the discussion part of the revised manuscript.

Comment 2:

Another important aspect is that VIP, as well as PACAP, are notorious for their vasodilator properties (PMID: 15959462, PMID: 15155712, PMID: 24563332), which may be prevented by administration of available VIP antagonists (PMID: 2116730). As such, it cannot be excluded that VIPHyb injections prevent local vasodilation and immune cells infiltration caused by LL37 in mutant mice. These considerations are important to determine a causal link with VIP induction in these animals.

Response 2:

We thank the reviewer for the helpful and constructive comments. We agreed that it cannot be excluded that VIPHyb injections prevent local vasodilation and immune cells infiltration caused by LL37 in mutant mice considering these aspects. To explore this possibility, we first detected the expression levels of VIP and PACAP in the local skin lesions, and found that VIP and PACAP were not affected in the skin lesions of mutant mice after LL37 administration (**shown in new Supplementary Figure 5c**); moreover, we subcutaneously injected with VIPHyb in LL37-induced wild type (WT) mice, and found that VIPHyb injections could not significantly alleviate rosacea-like phenotypes in WT mice (**shown in new Supplementary Figure 6**). Collectively, these new data

provided in the revised manuscript support the notion that VIPHyb injections do not alleviate the rosacea-like symptoms by preventing local vasodilation and immune cells infiltration caused by LL37 in mutant mice.

Comment 3:

Other less important remarks relate to the use of PGP9.5 as a neural marker. PGP9.5 acts as a tissue-specific ubiquitin carboxyl terminal hydrolase isoenzyme, and is also known as UCH-L1. It is abundant in peripheral nerves, and broadly used to detect intra-epidermal nerve fibres (PMID: 33892490, PMID: 35295465, PMID: 28914336). Whilst this makes PGP9.5 a suitable marker to co-localise nerve terminals in the skin, they should cautiously refer to it as a marker of intra-dermal nerve fibres rather than neural marker.

Response 3:

We thank the reviewer for the kind suggestion. As suggested, we have indicated PGP9.5 as a marker of intra-dermal nerve fibres rather than neural marker in the revised manuscript, which will not affect the conclusions generated from the related data.

Comment 4:

Finally, HCN-2 cortical neuron cell line (ATCC, CRL-10742) undergo senescence relatively quickly (> passage 21), so it is important that the authors indicate the passage number in the methods.

Response 4:

We thank the reviewer for the kind suggestion. The HCN2 cortical neuron cell line (passage 13) was obtained from ATCC, and the cells were expanded and were used at passage 16-17. As suggested, we have also indicated the passage number in the methods of the revised manuscript.

REVIEWERS' COMMENTS

Reviewer #1 (Remarks to the Author):

The revised manuscript is much improved and very well described. However, concerns remain regarding the diagnosis process. The reviewer recommends that authors analyze the group diagnosed only by photographs and telephone consultation separately from those diagnosed by in-person consultation.

Reviewer #2 (Remarks to the Author):

The authors have answered all my questions and suggestions and added them to the revised version of their paper.

The revised paper has an excellent quality and great novelty.

I recommend to accept the revised version of the manuscript for publication.

Reviewer #3 (Remarks to the Author):

This article is much improved and the topic is a very worthy one. Clearly the authors have done a lot of work. Some issues remain that require clarification:

1) The authors show multiple families in the validation groups with two affected individuals in each family harboring the same mutation. Were these mutation heterozygous or homozygous? Also it would be more believable if there were multiple siblings with the phenotype and genotypes.

2) The Discussion section needs tightening and word-smithing as some of the verbiage is not clear

3) What percent of genetic rosacea is explained by the variants you found? I know the authors state these variants are rare, but are we talking about <1% or <5%? This information helps to place the work in context

4) There are multiple typos in figure on PDF page 44

5) There are multiple typos in the figure on PDF on page 30

Reviewer #4 (Remarks to the Author):

Dear Editor,

The present work provides important indications about the genetic contribution of specific single gene mutations (LRRC4, SH3PXD2A and SLC26A8) in the development of rosacea. In the revised manuscript, the authors addressed the most critical concerns raised by this and other reviewers and where requested, conducted additional experiments (i.e. bulk RNA-seq) to provide additional evidence of the likely contribution of neuronal-derived VIP (and not PACAP) to disease pathogenesis. The authors also highlight the potential limitations of using VIPHybrid due to its lack of pharmacological specificity as a VIP antagonist, considering the additional antagonistic activity on other PACAP/VIP receptors (PAC1, VPAC1 and VPAC2). In conclusion, the revised work now convincingly show that the observed genetic mutations interfere (at least indirectly) with neuronal VIP expression at the peripheral nerve terminals reaching skin lesions, displaying a key involvement in rosacea pathogenesis.

This reviewer is satisfied by the quality of the revised work and has no further comments/criticism to raise.

Reviewer #1

Comments:

The revised manuscript is much improved and very well described. However, concerns remain regarding the diagnosis process. The reviewer recommends that authors analyze the group diagnosed only by photographs and telephone consultation separately from those diagnosed by in-person consultation.

Response:

We thank the reviewer for the recognition of our work and kind suggestions. In fact, for individuals whose blood samples were collected, the diagnosis was performed in person visits by 2 experienced dermatologists from the department of dermatology, Xiangya Hospital of Central South University, which was also indicated in the Methods of the manuscript. It means that all individuals whose blood samples were performed with WGS or WES or Sanger sequencing, were diagnosed by in-person consultation. So, in the process of data analysis, there is no group diagnosed only by photographs and telephone consultation.

Reviewer #2

Comments:

The authors have answered all my questions and suggestions and added them to the revised version of their paper.

The revised paper has an excellent quality and great novelty.

I recommend to accept the revised version of the manuscript for publication.

Response:

We thank the reviewer for the recognition of our work.

Reviewer #3

Overall comments:

This article is much improved and the topic is a very worthy one. Clearly the

authors have done a lot of work. Some issues remain that require clarification:

Overall response:

We thank the reviewer for the recognition of our work, and the kind suggestions, all of which have been addressed, as detailed below.

Comment 1:

1) The authors show multiple families in the validation groups with two affected individuals in each family harboring the same mutation. Were these mutation heterozygous or homozygous? Also it would be more believable if there were multiple siblings with the phenotype and genotypes.

Response 1:

We thank the reviewer for the kind suggestions. These mutations were heterozygous, which have been described in the Result section as suggested. We have listed all the siblings and their phenotype in all validation families (Shown in **Supplementary Figure 2**).

Comment 2:

2) The Discussion section needs tightening and word-smithing as some of the verbiage is not clear

Response 2:

We thank the reviewer for the kind suggestion. We have checked and modified the expression throughout Discussion section as suggested.

Comment 3:

3) What percent of genetic rosacea is explained by the variants you found? I know the authors state these variants are rare, but are we talking about <1% or <5%? This information helps to place the work in context.

Response 3:

We thank the reviewer for the kind suggestion. Totally, 14 out of 52 (27%) families had been identified with the variants in SH3PXD2A, SLC26A8 and LRR

family genes (Shown in **Supplementary Data 1** and **Supplementary Data 2**). We selected variants with minor allele frequency of < 0.01 , and we have added this information in the Result section as suggested, which was also indicated in **Figure 1**.

Comment 4:

4) There are multiple typos in figure on PDF page 44.

Response 4:

We thank the reviewer for the kind suggestion. We have corrected the typos in figure on PDF page 44 as suggested.

Comment 5:

5) There are multiple typos in the figure on PDF on page 30.

Response 5:

We thank the reviewer for the kind suggestion. We have corrected the typos in figure on PDF page 30 as suggested.

Reviewer #4

Comments:

The present work provides important indications about the genetic contribution of specific single gene mutations (LRRC4, SH3PXD2A and SLC26A8) in the development of rosacea. In the revised manuscript, the authors addressed the most critical concerns raised by this and other reviewers and where requested, conducted additional experiments (i.e. bulk RNA-seq) to provide additional evidence of the likely contribution of neuronal-derived VIP (and not PACAP) to disease pathogenesis. The authors also highlight the potential limitations of using VIPHybrid due to its lack of pharmacological specificity as a VIP antagonist, considering the additional antagonistic activity on other PACAP/VIP receptors (PAC1, VPAC1 and VPAC2). In conclusion, the revised work now convincingly show that the observed genetic mutations interfere (at least

indirectly) with neuronal VIP expression at the peripheral nerve terminals reaching skin lesions, displaying a key involvement in rosacea pathogenesis.

This reviewer is satisfied by the quality of the revised work and has no further comments/criticism to raise.

Response:

We thank the reviewer for the recognition of our work.